# Incidence and Outcomes of Abdominal Aortic Aneurysm Repair in New Zealand from 2001 to 2021

**DOI:** 10.3390/jcm12062331

**Published:** 2023-03-16

**Authors:** Sinead Gormley, Oliver Bernau, William Xu, Peter Sandiford, Manar Khashram

**Affiliations:** 1Department of Vascular & Endovascular Surgery, Waikato Hospital, Hamilton 3204, New Zealand; 2Faculty of Medical & Health Sciences, University of Auckland, Auckland 1010, New Zealand; 3Planning Funding and Outcomes Unit, Auckland and Waitemata District Health Boards, Auckland 1010, New Zealand; 4School of Population Health, University of Auckland, Auckland 1010, New Zealand

**Keywords:** abdominal aortic aneurysm, incidence, epidemiology, EVAR, ruptured AAA, New Zealand

## Abstract

Purpose: The burden of abdominal aortic aneurysms (AAA) has changed in the last 20 years but is still considered to be a major cause of cardiovascular mortality. The introduction of endovascular aortic repair (EVAR) and improved peri-operative care has resulted in a steady improvement in both outcomes and long-term survival. The objective of this study was to identify the burden of AAA disease by analysing AAA-related hospitalisations and deaths. Methodology: All AAA-related hospitalisations in NZ from January 2001 to December 2021 were identified from the National Minimum Dataset, and mortality data were obtained from the NZ Mortality Collection dataset from January 2001 to December 2018. Data was analysed for patient characteristics including deprivation index, repair methods and 30-day outcomes. Results: From 2001 to 2021, 14,436 patients with an intact AAA were identified with a mean age of 75.1 years (SD 9.7 years), and 4100 (28%) were females. From 2001 to 2018, there were 5000 ruptured AAA with a mean age of 77.8 (SD 9.4), and 1676 (33%) were females. The rate of hospitalisations related to AAA has decreased from 43.7 per 100,000 in 2001 to 15.4 per 100,000 in 2018. There was a higher proportion of rupture AAA in patients living in more deprived areas. The use of EVAR for intact AAA repair has increased from 18.1% in 2001 to 64.3% in 2021. The proportion of octogenarians undergoing intact AAA repair has increased from 16.2% in 2001 to 28.4% in 2021. The 30-day mortality for intact AAA repair has declined from 5.8% in 2001 to 1.7% in 2021; however, it has remained unchanged for ruptured AAA repair at 31.6% across the same period. Conclusions: This study highlights that the incidence of AAA has declined in the last two decades. The mortality has improved for patients who had a planned repair. Understanding the contemporary burden of AAA is paramount to improve access to health, reduce variation in outcomes and promote surgical quality improvement.

## 1. Introduction

The epidemiology and management of Abdominal Aortic Aneurysm (AAA) has changed in the last 20 years [1]. Some of these include a decline in prevalence, improvement in life expectancy and the introduction of Endovascular Aneurysm Repair (EVAR), along with the improvements in pre-operative work up and peri-operative care and the establishment of AAA screening programmes in some countries. The introduction of EVAR has made it possible to offer AAA repair to patients that previously might not have been candidates for open aneurysm repair (OAR). As a result, the proportion of AAA repairs using EVAR has increased [2], and the survival of both intact and ruptured AAA has improved.

The approach to population level screening is highly variable between different countries. New Zealand (NZ) has not established a policy for national AAA screening and mostly relies on background detection from medical imaging despite demonstrated cost-effectiveness [3]. In the United Kingdom [4] and Sweden [5], population screening programmes have contributed to the changes seen in the contemporary management and outcomes of AAA, such as the reduced incidence of ruptured aneurysms.

The incidence of hospitalisations and mortality of AAA in NZ has been previously reported from the years 1994–2009 [6]. However, the rate of decline, the changes in AAA management and patient outcomes have not been reported. The objective of this study was to report the incidence and outcomes of AAA in NZ.

## 2. Materials and Methods

### 2.1. Study Design

This was a retrospective observational cohort study and was prepared according to the Strengthening the Reporting of Observational Studies in Epidemiology (STROBE) guidelines [7]. Ethics approval was granted by the New Zealand Health and Disability Ethics Committees (Re13/STH/190/AM01).

### 2.2. Study Protocol and Data Collection

There were three datasets that contributed to the study population. The Analytics Services Team from the New Zealand Ministry of Health provided admission data from the National Minimum Dataset (NMDS) for all publicly and privately funded hospitalisations with any diagnosis or procedure relating to AAA (Appendix A, Table A1), with a discharge date from 1 January 2001 to 31 December 2021. Operative data from January 2010 onwards were cross referenced against the Australasian Vascular Audit (AVA) [8,9,10]. All deaths registered in NZ are recorded on the National Mortality collection dataset, and this database was interrogated to retrieve all deaths with a primary diagnosis of aortic aneurysm (ICD-10-AM codes I71.3 and I71.4) from January 2001 to December 2018. This permitted defining aneurysm-related mortality for patients who died because of a AAA-specific cause. In addition, two further groups were created for patients who died with a ruptured AAA in the community or those presented to the emergency department who died prior to hospital admission, and for the patients who were admitted to hospital with a ruptured AAA but did not have a repair.

AAA-related hospitalisations and repairs were analysed from 2001 to 2021, and AAA-related deaths were analysed from 2001 to 2018, as there is an approximately three-year lag time. Data collected from the NMDS included baseline characteristics, diagnoses and procedures performed. Only patients with a AAA diagnosis (I71.3, I71.4) were analysed. Data are presented on a patient-level basis using the index presentation. For patients with multiple hospitalisations across the study period, the index presentation was defined as the first occurrence of an aneurysm-related procedure or rupture, or the initial presentation with a AAA diagnosis if neither procedure nor rupture was found. There was no look back period during the study period. A pre-hospital death was defined as a death occurring in the community or whilst in the emergency department prior to admission. Aneurysm-related mortality was defined as any death occurring within 30 days of aneurysm treatment or date of rupture, or any death with a primary cause relating to AAA.

### 2.3. Ethnicity Definition

The New Zealand Ministry of Health ethnicity data protocols dictate the use of prioritisation of ethnicities. This means that if a patient identifies with more than one ethnicity, specific protocols are put in place to determine which ethnic group a patient will be included in for the purposes of statistical analysis. This is designed to ensure indigenous communities are counted and prioritised. It also works to ensure other ethnic minorities are enabled with the largest possible inclusion of membership to enable appropriate statistical analysis to be undertaken. New Zealand national ethnicity standards encourage all primary, secondary and tertiary health institutions to have patients complete a form in which they can self-identify with the ethnic group or groups that best describe their ethnic affiliations [11].

### 2.4. New Zealand Index of Deprivation (NZDep)

The New Zealand Index of Deprivation (NZDep) is a measure obtained from census data and is linked to geographical location rather than individuals [12]. The NZDep was calculated based on nine domains: access to transport, access to communication, living space, income, recipient of benefit, single-parent family, home ownership, qualifications and employment. NZDep groups deprivation scores into deciles, with 1 being the least deprived areas and 10 being the most deprived areas [13].

### 2.5. AAA-Related Hospitalisations

Age- and sex-specific rates per 100,000 population per year were calculated from the NZ population at each respective year. The World Health Organisation (WHO) standard population was used to age standardise the rates. All cases identified were assumed to be new cases and each case was identified once only.

### 2.6. Statistical Analysis

Data were presented as frequency (percentage) for categorical data and mean ± standard deviation (SD) or median (interquartile range). The Chi-squared test was used to compare categorial data and Student’s *t*-test (two-tailed) was used for continuous variables. Age standardisation was completed using standard populations modelled after the NZ World Health Organisation standard population as per the methods of Robson et al. [14]. Age standardisation was completed with the dsr package, and 95% confidence intervals were calculated using well-documented methods. Statistical analyses were completed in R version 3.6.1 (R foundation for statistical computing, Vienna, Austria) [15].

## 3. Results

### 3.1. Patient Demographics

From January 2001 to December 2021, 14,436 patients were diagnosed with an intact AAA or registered as having died from an intact AAA-related death in NZ. The mean age was 75.1 years (SD 9.7 years), and 4100 (28%) were females. From January 2001 to December 2018, 5000 patients presented or died with a diagnosed ruptured AAA (rAAA). The mean age was 77.8 years (SD 9.4 years), and 1676 (33%) were females. For both intact and ruptured AAA groups, NZ Europeans made up over 80% of each patient cohort. There was a higher proportion of ruptured and intact AAA in those living in more deprived areas, with 4009 (22.7%) of patients living in deprivation deciles 9 to 10 versus 1993 (11.3%) in deciles 1 to 2. Patient characteristics are described for intact and ruptured AAA index presentations stratified by sex in Table 1. AAA prevalence grouped by age and sex are described in Figure 1. Crude estimated incidence of AAAs per 100,000 person-years stratified by gender and age are described in Table 2.

For the period 2001–2018, a total of 3874 AAA-related deaths were documented. There were 3601/3874 (93.0%) deaths from ruptured AAA and 273/3874 (7.0%) from intact AAA. The overall mortality for rAAA patients that underwent EVAR was 17/94 (18.1%). The overall mortality for rAAA patients that underwent OAR was 495/1511 (32.8%). Overall, there was a decline in AAA-related deaths per year across the study period, with 249 per year from 2001 to 2005 and 179 per year from 2014 to 2018.

### 3.2. AAA Presentation and Repair

From 2001 to 2018, there was a 64.0% decrease in AAA presentations in males, from 61.5 to 22.1 per 100,000 per year. This was also observed in female presentations. There has been a decline in the incidence of intact AAA presentations and AAA repair from 24.8 per 100,000 in 2001 to 12.6 per 100,000 in 2021 and 11.9 per 100,000 in 2001 to 7.5 per 100,000 in 2021, respectively. There was a greater decline in the incidence of AAA hospitalisations in Maori females versus Maori males between 2001 and 2018. In 2018, there was 21.3 per 100,000 presentations in Maori females versus 28.3 per 100,000 in Maori males (Figure 2).

For ruptured AAA, the incidence decreased from 9.7 in 2001 to 4.7 per 100,000 in 2018. Similarly, repairs of ruptured AAA reduced from 4.0 per 100,000 in 2001 to 1.4 per 100,000 in 2018 (Figure 3).

### 3.3. Trends of AAA Repair and Presentation

From 2001 to 2021, considering the AAA repairs performed, a total of 8421 (82.4%) were on intact AAA and 1802 (17.6%) were on ruptured AAA. The average annual number of intact AAA repairs remained fairly constant during the study period until 2015 after which it declined steadily.

From 2001 to 2018, considering those with a diagnosis of ruptured aneurysm, 1605 (32.1%) had a repair, 1673 (33.5%) were palliated or denied surgical intervention (treated conservatively) in the hospital and 1722 (34.4%) died prior to hospitalisation. Over the period 2001–2018, there was a decrease in the rate of patients with ruptured AAA dying pre-hospital from 4.2 per 100,000 population to 1.4 per 100,000 (Figure 3).

### 3.4. AAA Repair in Octogenarians

We observed a significant shift in AAA presentation to the older population driven by both the increased use of EVAR and the increasing number of octogenarians. The proportion of octogenarians undergoing intact AAA repair has increased from 16.2% in 2001 to 28.4% in 2021 (Figure 4). We also observed a decline in 30-day post-operative mortality in this age group following AAA repair, from 6.8% in 2001 to 0.9% in 2021.

### 3.5. Incidence of Ruptured AAA Stratified by Sex

From 2001 to 2018, there was a total of 5000 ruptured AAA presentations, of which 3324 (66.5%) were males and 1605 (32.1%) underwent repair. A greater number of males underwent surgery for ruptured AAA (1266 (78.9%) males). The proportion of males with a non-operative ruptured AAA decreased. The number of females with a ruptured AAA turned down for repair declined by 50% (Figure 5).

### 3.6. Methods of AAA Repair and Trends in Operative Mortality

EVAR has gradually replaced OAR for patients requiring an intact AAA repair (4439 (52.7%) OAR cases vs. 3982 (47.2%) EVAR cases). There was an increase in the use of EVAR in all age groups for intact AAA repair from 18.1% in 2001 to 64.3% in 2021. There was a decline in 30-day mortality for patients undergoing intact AAA repair from 5.8% in 2001 to 1.7% in 2021. This coincided with the rise of EVAR usage from 18.1% in 2001 to 64.3% in 2021 (Figure 6).

For patients undergoing ruptured AAA repair, there was no change in outcomes with a mean 30-day mortality of 31.6%. There was an increase in EVAR usage for ruptured AAA from 0.8% in 2001 to 28.8% in 2021. (Figure 7). The 30-day mortality for patients who had an EVAR for a ruptured AAA (rEVAR) decreased from 34.1% to 28.8%. There was a decline in 30-day mortality for OAR from 5.4% in 2001–2005 to 4.2% in 2016–2020 (Figure 8).

### 3.7. Effect of Centralisation on Intact AAA Repair

Since centralisation of vascular surgical services has been implemented in NZ, there has been a steady increase in intact AAA repairs performed by tertiary centres since 2012, as demonstrated in the Figure 9. This increased from 79.2% in 2001 to 89.1% in 2021. As a result, AAA repairs are no longer performed in level three and four centres.

## 4. Discussion

The salient findings observed in this study were firstly that the outcomes of intact AAA repair have improved during the last decade, with an overall reduction in 30-day mortality risk. Second, the overall counts of AAA repairs have remained fairly steady, but the age-standardized incidence has declined. However, the proportion of octogenarians undergoing intact AAA repair has increased with a significant reduction in mortality. Third, the incidence of ruptured aneurysms has not changed. In describing these findings, the disparity of sex on aneurysm presentations and outcomes has also become more apparent. These observations support that the burden of AAA disease in NZ that requires further investigation is likely to continue with the aging population and improved life expectancy.

This study differs from previous epidemiological studies of AAA in that we separated the AAA presentations into categories of acuity and repair [6,16]. By doing this, we noted that the largest decline in AAA presentations was in patients who had a had an intact AAA but did not have surgery. This group most likely presents patients being hospitalised for a non-AAA-related admission but who had an AAA present. In addition, we had access to patient unique identifiers; therefore, each hospitalisation was counted once, and the most clinically relevant AAA hospitalisation was identified.

### 4.1. Incidence of AAA

Sandiford and colleagues reported that AAA incidence, mortality, hospital admissions and hospital death rates between 1995 and 2008 in NZ have declined. In contrast to Sandiford’s report, AAA presentations in this study were separated in order to provide some explanation for this decrease in age-standardized incidence [6]. In doing so, one of the major contributors to this decline appeared to be those patients who had an AAA diagnosis but did not have a repair. In 2001, there was 19.41 per 100,000 cases of intact AAA not repaired and in 2021 there was 4.94 cases per 100,000. In Sandiford’s paper, AAA presentations were divided into ruptured and non-ruptured AAAs. In 2001, there was 5.29 per 100,000 cases of ruptured AAAs and 25.99 per 100,000 cases of unruptured AAAs.

In the 1990s, studies reported an increase in the incidence of asymptomatic AAA [17]. Smoking is considered one of the most causative risk factors for developing an AAA, and the decline in smoking prevalence in developed countries over the past 30 years is considered an important reason for the decline in the disease burden of AAA [18]. Norman et al. reported a declining rate of hospitalisation for both ruptured and non-ruptured AAA with a 38% decline in AAA mortality in men from 1999 to 2006 in Australia [19], and similarly, Sandiford et al. reported a 53% reduction in mortality from 1991 to 2007 in NZ [6]. In our study, we noted a decline in AAA incidence to be most prominent between 2005 and 2007.

The overall hospital presentations with ruptured and intact aneurysms have declined. As hospital admissions for AAA have decreased, the operative intervention for AAA has also declined due to the reduction in open aortic repair despite the increase in EVAR utilisation. Similar operative trends have also been observed in the UK National Vascular Registry and in the Swedish Vascular Registry [20,21]. These trends could be explained by the substantial changes that have influenced AAA management over the last two decades including the establishment of EVAR, the implementation of cardiovascular risk factor management, the introduction of statins and increased public health awareness on the importance of lowering blood pressure and smoking cessation. In addition, advances in perioperative medicine and quality improvement initiatives, such as centralisation of vascular surgery services, may have influenced these trends.

### 4.2. Effect of Social Deprivation and Ethnicity on Incidence of AAA

There is a higher proportion of intact and rupture AAA in patients living in more deprived areas in New Zealand. Socio-economic status (SES) and ethnicity have been reported as markers influencing the likelihood of increased mortality. A New Zealand study investigated how these factors impacted patient survival after AAA repair over a 14.5 year period and observed that patients living in areas of higher social deprivation had a higher risk of short- and medium-term mortality after AAA repair in a universal health setting [22]. The greatest proportion of Maori undergoing AAA repair lived in the most deprived areas, deciles 9–10. NZ Europeans were more likely to present to the hospital electively, live in less deprived areas and had the highest proportion undergoing an aneurysm repair at a private institution. In contrast the other ethnic groups, they had a higher proportion of patients presenting acutely with AAA rupture.

In a previous study to determine the prevalence of AAA in New Zealand, a group of patients undergoing computed tomography colonography had their infrarenal aorta measured [23]. The data suggest that the background detection of AAA might be lower in non-NZ Europeans, which might explain the higher rate of acute AAA hospital presentations. In addition, Maori men have a lower life expectancy in the general population than other men in New Zealand, resulting in an under-representation in those with AAA. The first population-based cross-sectional prevalence study to report the prevalence of AAA in NZ Maori or any other Polynesian group was published recently by Sandiford et al. Their study concluded that the prevalence of undiagnosed AAA in New Zealand Maori men is considerably higher than in screened populations of equivalent age in the United Kingdom and Sweden. The impact of ethnicity is likely to become more relevant given that the proportion of Maori patients has increased in the last 10 years and is likely to continue to do so. Our study has noted a change in population demographics over the last 20 years, and the proportion of non-NZ Europeans is increasing.

### 4.3. Mortality and Increase in EVAR Use

EVAR has resulted in a paradigm shift in the treatment of AAA worldwide and has gradually replaced open surgery [24]. In NZ, we have observed similar number of OAR and EVAR in the last 10 years. This can be partly explained by access to universal national healthcare and vascular surgeons’ familiarity and skill set with both procedures. The decline in 30-day mortality following intact AAA repair has been observed elsewhere and has been predominantly related to the rise in EVAR usage [19,25]. Improvements in pre-operative, peri-operative and post-operative care along with centralisation of vascular services may have also contributed to the lower operative mortality. Previous studies have demonstrated that the mortality from AAA has declined in the last two decades in several countries, but this has not been consistent in all regions of the world [26,27]. A recent study analysed the death burden of aortic aneurysm and trends of four risk factors from 1990 to 2019 using the updated 2019 Global Burden of Disease study database and discovered that the global burden of death attributable to AAA began to increase after decreasing for two decades [28]. The study suggests that this trend will continue for the next decade and that high systolic blood pressure will replace smoking as the most important risk factor associated with aortic aneurysm death. A meta-analysis has shown that the risk of rupture in AAA patients with comorbid hypertension was 1.66 times higher than that in patients without comorbid hypertension [29].

### 4.4. Octogenarians

Predictions estimate that the population aged over 80 will increase fivefold by 2040. The repair of intact and ruptured AAA has increased in the older population with octogenarians constituting a significant fraction of intact AAA repairs performed in several countries [30]. In Sweden, the incidence of AAA repair in octogenarians has nearly tripled from 13% to 36% per 100,000 population >80 years in Sweden over the periods 1994–1999 to 2010–2014 [5]. Similarly in our study, AAA repair in octogenarians has almost doubled from 16.2% in 2001 to 28.4% in 2021. In a study by Park et al. among octogenarians treated for an intact AAA, 80% were treated by EVAR and patients older than 80 accounted for 25% of the total EVAR cohort [31].

This shift in AAA presentation to the older population has been driven by the increased use of EVAR and an increasing number of active octogenarians. A recent meta-analysis by Sweeting et al. of the Ruptured Aneurysm Trialists research group demonstrated that the 1-year mortality rate for an octogenarian with a ruptured AAA were 35% (95% CI = 18–56) following EVAR and 54% (95% CI = 47–60) following OAR compared to the overall population, with 38.6% treated by EVAR and 42.8% treated by OAR [32]. Compared to younger patients, EVAR in octogenarians is associated with a significantly higher but still acceptable peri-operative and midterm mortality rate. In our study, the 30-day mortality of octogenarians undergoing intact AAA repair decreased from 6.8% in 2001 to 0.9% in 2021. A multi-centre retrospective study was carried out in the Netherlands investigating the outcomes of ruptured AAA in octogenarians [33]. After one year, half of the octogenarians operated on for a ruptured AAA were alive, with >80% living at home.

### 4.5. Ruptured AAA Outcomes

AAA-related mortality is estimated at 150,000–200,000 deaths per year worldwide, which is equivalent to various types of cancer [34]. On average, since 2001, in New Zealand, approximately 215 people per year are recorded as dying of ruptured AAA, of which 80.9% are the result of ruptured AAA without undergoing any form of repair, and the remaining are a consequence of undergoing AAA procedures predominately for ruptured aneurysms.

As a substantial proportion of patients with rAAA die before hospitalisation, most studies do include prehospital deaths. Our study demonstrated that the overall mortality of rAAA was 72%, which is similar to a Norwegian study that found the incidence to be 68% [35]. In addition, the mortality for patients undergoing EVAR for rAAA (rEVAR) is static, from 34.1% in 2001 to 28.8% in 2021. During the same period, we observed a significant increase in EVAR utilisation from < 1% to 28.8%. Historically, the mortality rate for patients with ruptured aneurysms who undergo open surgery is 41% to 49% [36]. Centres that report outcomes for all rAAAs after the introduction of rEVAR have published mortality rates varying between 24% and 46%, and our results are keeping in line with these other studies [37].

### 4.6. Effect of Centralisation of Aortic Pathology in NZ

The volume outcome relationship in aortic aneurysm surgery has been well-studied, demonstrating that higher volume centres produce the best patient outcomes, which has subsequently led to a drive for centralisation of aortic and complex endovascular surgery [38,39]. A meta-analysis of the literature in 2007, comprising 421,299 elective aneurysm repairs, reported a weighted odds ratio of 0.66 in favour of higher volume centres [40]. An example of the benefits of the centralisation of vascular surgical services has been observed in the United Kingdom (UK) [41]. Twenty years ago, outcomes from aortic aneurysm surgery in the UK were among the worse in the Western world. Patients in lower volume centres had a higher mortality rate and poorer access to endovascular treatment than those treated in a higher volume centre. This subsequently led to a national quality improvement programme in AAA surgery in the last decade, which has resulted in a significant improvement in outcomes of AAA repair. In NZ, the model of care that is supported for vascular services is a regional model. Services are organised around Level 5 and/or 6 specialist vascular centres that provide a comprehensive range of vascular and endovascular services for adults and include the following hospitals: Auckland City Hospital, Waikato Hospital, Wellington Hospital, Christchurch Hospital, Middlemore Hospital and Dunedin Hospital. There has been no formal centralisation of vascular surgical services implemented, but as evident from these data, there has been a steady increase in intact AAA repairs performed in tertiary centres in the last decades. Vascular surgery only separated from general surgery training in NZ between 1995 and 1997 with the establishment of the establishment of the Board of Vascular Surgery of the Royal Australasian College of Surgeons (RACS) In 2002, the Australian and New Zealand Society for Vascular Surgery (ANZSVS) became the official administrator of vascular surgery in ANZ in association with RACS [42].

### 4.7. Limitations

This study has several limitations. As with all administrative databases, the data used in this study are subject to coding errors. Patients who were not hospitalised with a diagnosis of AAA or died without diagnosis despite having a AAA, might have been missed in our data capture. We are also unable to report the number of intact AAAs treated conservatively as surgical decision making was not recorded in our data. The AVA was implemented at the end of 2010, so we only have records of procedures performed in in private hospitals over the last 12 years.

In addition, our database contains no information regarding why EVAR was chosen over OAR or conservative management. Administrative data were used rather than patient-level data, so the patient risk profile, AAA diameter and extent of AAA anatomy were not recorded. Death status was based on mortality records that include only those deaths that occurred within New Zealand. Any deaths occurring outside New Zealand would not be captured, and this may have resulted in some degree of under reporting. We are unable to report mortality data from 2018 to 2021, as there is a three-year lag time with obtaining these data from the national mortality collection database.

## 5. Conclusions

This study highlights the epidemiological trends and survival outcomes of AAA management in NZ over 20 years and the challenges health services might encounter from the AAA burden. Important trends include the stabilisation of intact AAA repair, an increase in the number of octogenarians with AAA disease and the mortality rate of rAAA, which has remained static. Understanding the changing pattern of AAA burden is paramount to improve resource allocation and promote surgical quality improvement.

## Figures and Tables

**Figure 1 jcm-12-02331-f001:**
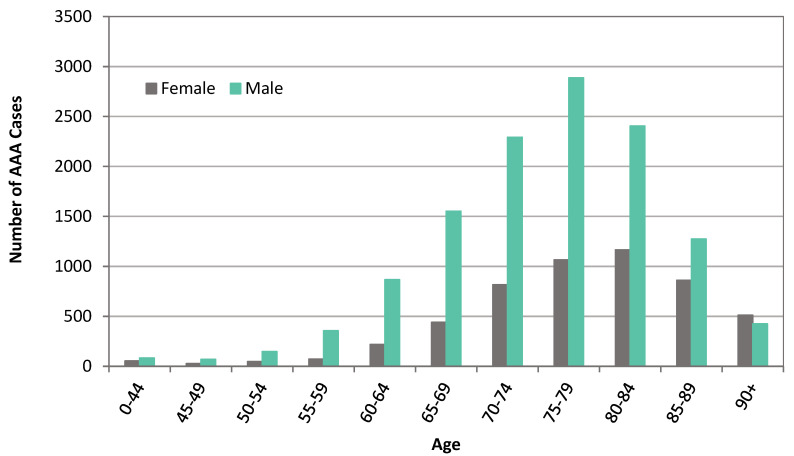
AAA prevalence grouped by age and sex, 2001–2018.

**Figure 2 jcm-12-02331-f002:**
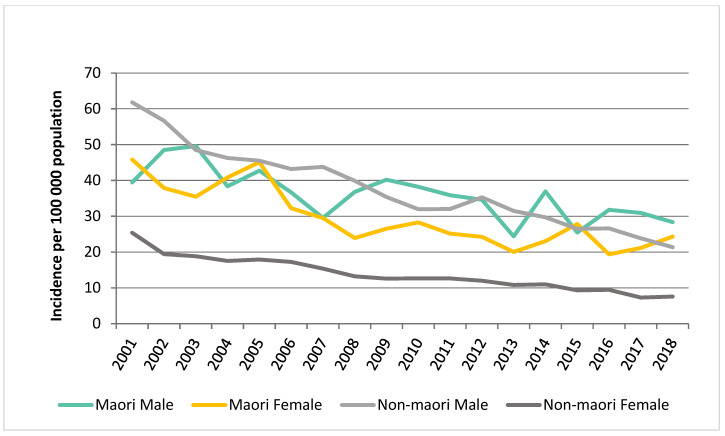
AAA hospitalisation age standardized to NZ population changes.

**Figure 3 jcm-12-02331-f003:**
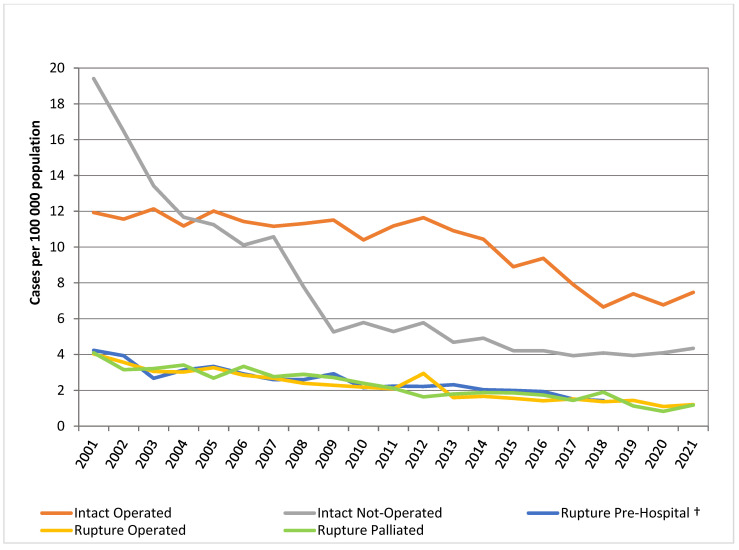
Trends in AAA presentation and management. ^†^ Pre-hospital rupture data available only from 2001 to 2018.

**Figure 4 jcm-12-02331-f004:**
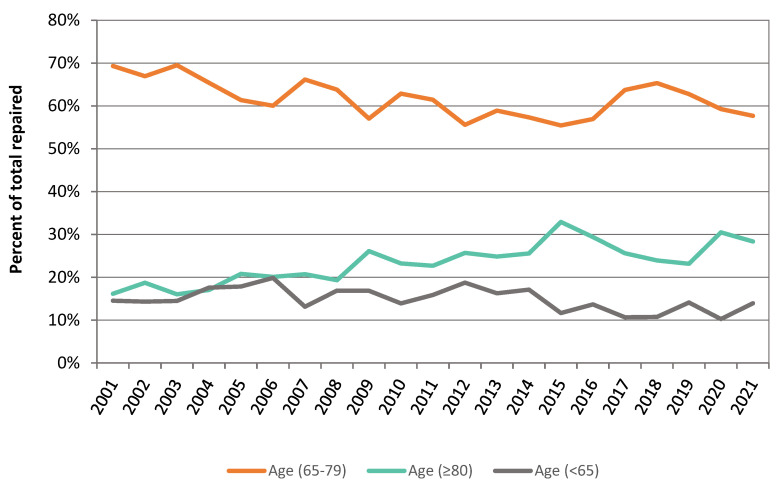
Proportion of intact AAA repaired by age.

**Figure 5 jcm-12-02331-f005:**
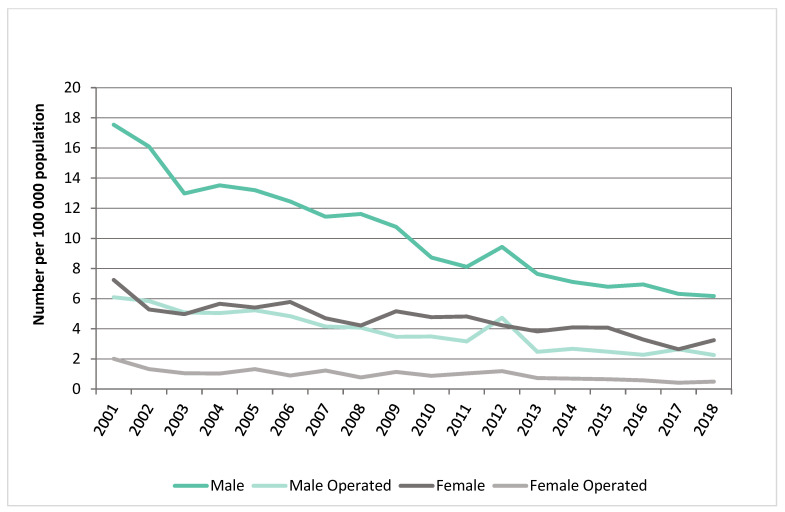
Trends or ruptured AAA incidence and repair between sexes.

**Figure 6 jcm-12-02331-f006:**
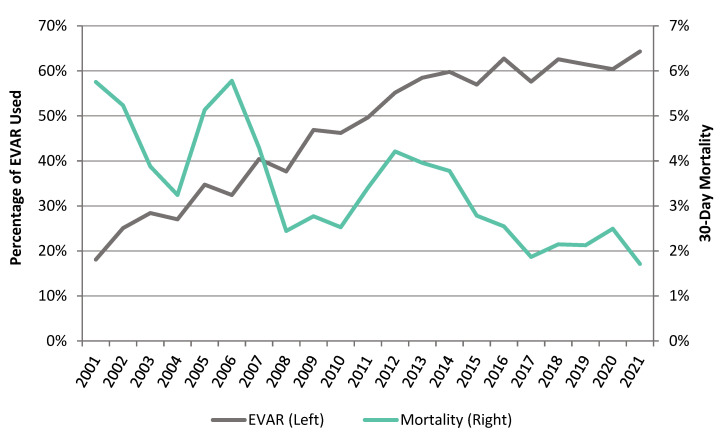
Intact AAA 30-day post-operative mortality and proportion of EVAR usage.

**Figure 7 jcm-12-02331-f007:**
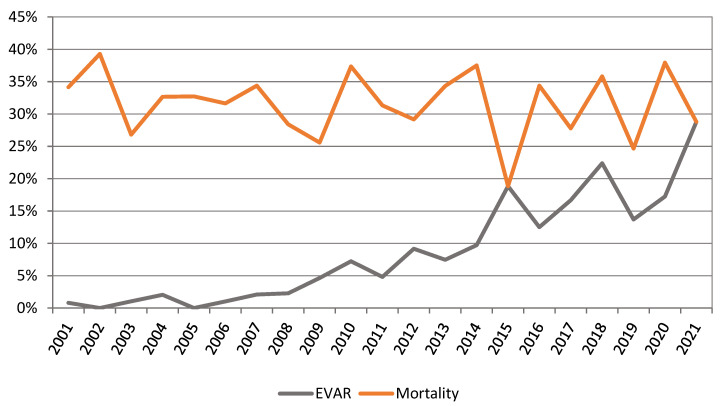
Ruptured AAA 30-day mortality versus proportion of EVAR used.

**Figure 8 jcm-12-02331-f008:**
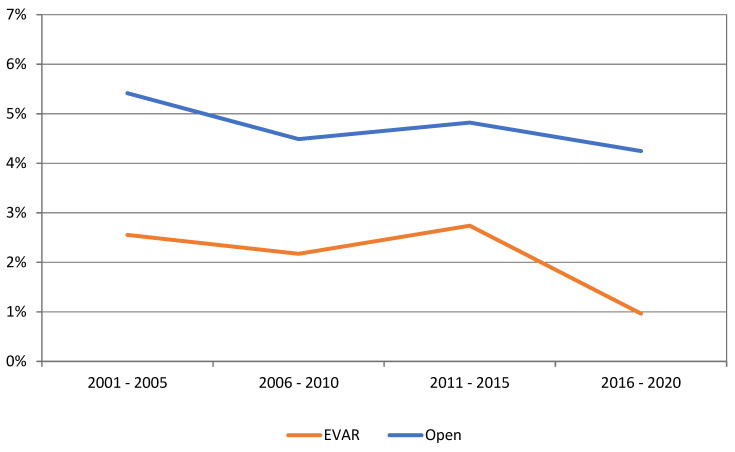
Intact AAA 30-day mortality by repair type.

**Figure 9 jcm-12-02331-f009:**
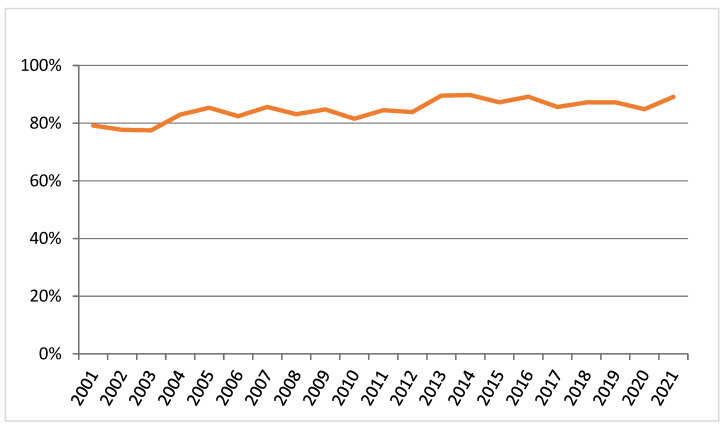
Trends of intact AAA repair performed by tertiary vascular centres.

**Table 1 jcm-12-02331-t001:** Patient characteristics for intact and ruptured AAA index presentations stratified by sex. Intact AAA data from 2001 to 2021 and ruptured AAA data from 2001 to 2018.

	Intact (*n* = 14,436)			Ruptured (*n* = 5000)		
	Male	Female	*p*-Value	Male	Female	*p*-Value
**Number**	10,336	4100		3324	1676	
Age, mean (SD)	74.5	76.8	<0.001	76.5	80.6	<0.001
Ethnicity						
NZ/Other European	8794 (85.1%)	3319 (81%)	<0.001	2853 (85.8%)	1402 (83.7%)	<0.001
Māori	614 (5.9%)	480 (11.7%)		232 (7.0%)	176 (10.5%)	
Pacific	240 (2.3%)	84 (2.0%)		75 (2.3%)	40 (2.4%)	
Asian, African, Hispanic	295 (2.9%)	100 (2.4%)		80 (2.4%)	40 (2.4%)	
Not specified	393 (3.8%)	117 (2.9%)		84 (2.5%)	18 (1.1%)	
Deprivation index						
1–2	1371 (13.3%)	452 (11%)	<0.001	308 (9.3%)	122 (7.3%)	<0.001
3–4	1687 (16.3%)	601 (14.7%)		410 (12.3%)	199 (11.9%)	
5–6	2129 (20.6%)	867 (21.1%)		520 (15.6%)	235 (14%)	
7–8	2557 (24.7%)	1060 (25.9%)		631 (19%)	304 (18.1%)	
9–10	2349 (22.7%)	1058 (25.8%)		665 (20%)	342 (20.4%)	
Not available	243 (2.4%)	62 (1.5%)		790 (23.8%)	474 (28.3%)	
Intervention						
OAR	3411 (33%)	1028 (25.1%)	<0.001	1190 (35.8%)	321 (19.2%)	<0.001
EVAR	3150 (30.5%)	832 (20.3%)		76 (2.3%)	18 (1.1%)	
Not operated	3775 (36.5%)	2240 (54.6%)	<0.001	2058 (61.9%)	1337 (79.8%)	<0.001

**Table 2 jcm-12-02331-t002:** Crude estimated incidence of AAAs/100,000 person-years using NZ census statistics stratified by gender and age.

Category	Cases	Crude Overall Incidence (95% CI)
**Gender**		
Female	5905	12.44 (12.13–12.76)
Male	13929	30.34 (29.84–30.85)
**Age Group**		
44 Years and Under	300	0.04 (0.04–0.05)
45–49 Years	210	0.27 (0.23–0.31)
50–54 Years	432	0.59 (0.53–0.64)
55–59 Years	1016	1.52 (1.43–1.62)
60–64 Years	2424	4.22 (4.06–4.39)
65–69 Years	4530	9.49 (9.22–9.77)
70–74 Years	7038	18.44 (18.01–18.88)
75–79 Years	8788	30.46 (29.38–31.1)
80–84 Years	8024	39.8 (38.93–40.68)
85–89 Years	4780	40.98 (39.83–42.16)
90 Years and Over	2126	35.66 (34.16–37.2)

## Data Availability

Ethics approval was granted by the New Zealand Health and Disability Ethics Committees (Re13/STH/190/AM01).

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
