# Peer review of "Incidence and Outcomes of Abdominal Aortic Aneurysm Repair in New Zealand from 2001 to 2021"

_jcm, 2023, doi:10.3390/jcm12062331_

Round 1
Reviewer 1 Report
The study is well written. There are only some comments.
Line 24 “There was a higher proportion of rupture AAA in patients living 24 in higher deprived areas.” Is there an explanation for this result?
Line 122: “For both intact and ruptured AAA groups, NZ Europeans made up over 80% of each patient cohort.” Can you please explain and justify these results in the discussion?
Line 139-141: “There 139 has been a decline in the incidence of intact AAA presentations and AAA repair from 24.8 140 per 100,000 in 2001 to 12.6 per 100,000 in 2021 and 11.9 per 100,000 in 2001 to 7.5 per 100,000 141 in 2021, respectively.”
How would you explain the reasons for the decrease in the incidence of intact and ruptured AAA?
Line 237: please remove “a had”.
Author Response
Comments and Suggestions for Authors:
Line 24 “There was a higher proportion of rupture AAA in patients living 24 in higher deprived areas.” Is there an explanation for this result?
Thank you for your comment. There is a higher proportion of rupture AAA in patients living in higher deprived areas in New Zealand.
Socio-economic status (SES) and ethnicity have been reported as markers influencing the likelihood of increased mortality. In New Zealand, Maori, the indigenous people, constitute 15% of the total population and Asian and Pacific Island people constitute 12% and 7% of the total population respectively. A New Zealand study by Khashram et al investigated how socioeconomic status and ethnicity impacted patient survival after abdominal aortic aneurysm (AAA) repair. [1] Over a 14.5 year period they observed that patients living in areas of higher social deprivation had a higher risk of short and medium-term mortality after AAA repair in a universal health setting. In addition, those who identified as Maori, had a higher overall medium-term mortality despite similar short-term outcomes compared with other ethnic groups. Living in areas of high social deprivation (deprivation decile 7 or greater) was an independent predictor of short and medium-term overall mortality when compared with living in deprivation deciles 1 or 2. The greatest proportion of Maori undergoing AAA repair lived in the highest deprived areas, deciles 9-10. NZ Europeans were more likely to present to hospital electively, live in less deprived areas, and had the highest proportion undergoing an aneurysm repair at a private institution. In contrast the other ethnic groups had a higher proportion of patients presenting acutely with AAA rupture.
Previous studies have also reported higher AAA mortality in Maori than in other ethnic groups [2] Maori tended to present with AAA at a younger age than non-Maori. Sandiford et al have supported these findings and also demonstrated higher AAA mortality among Maori and Pacific Island men, which was partially attributed to a higher proportion of non-elective repairs than in NZ Europeans.[3]
Line 122: “For both intact and ruptured AAA groups, NZ Europeans made up over 80% of each patient cohort.” Can you please explain and justify these results in the discussion?
Thank you for your comment. Our results support that NZ Europeans do make up 80% of each patient cohort for both intact and ruptured AAA groups.
In a previous study to determine the prevalence of AAA in New Zealand , a group of patients undergoing computed tomography colonography had their infrarenal aorta measured. They identified that the proportion of Maori was only 2%, which is less than the 6%-7% expected population in this region. The data suggests that the background detection of AAA might be lower in non-NZ Europeans, which might explain the higher rate of acute AAA hospital presentations.[4] In addition Maori men have a lower life expectancy in the general population than other men in New Zealand, resulting in an under-representation in those with AAA.
The first population based cross sectional prevalence study to report the prevalence of AAA in NZ Maori or any other Polynesian group was published recently by Sandiford et al. Their study concluded that the prevalence of undiagnosed AAA in New Zealand Māori men is considerably higher than in screened populations of equivalent age in the United Kingdom and Sweden. Maori women in particular have been reported to experience significantly higher hospitalization rates for AAA.[5]
The impact of ethnicity is likely to become more relevant given that the proportion of Maori patients has increased in the last 10 years and is likely to continue to do so. Our study has noted a change in population demographics over the last 20 years and the proportion of non NZ Europeans is increasing.
Line 139-141: “There has been a decline in the incidence of intact AAA presentations and AAA repair from 24.8 140 per 100,000 in 2001 to 12.6 per 100,000 in 2021 and 11.9 per 100,000 in 2001 to 7.5 per 100,000 141 in 2021, respectively.”
How would you explain the reasons for the decrease in the incidence of intact and ruptured AAA?
Thank you for your comment.
Aneurysm related mortality from ruptured and intact AAA has gradually declined over the last 20 years for males and females. The introduction of endovascular aneurysm repair (EVAR) for the management of aneurysmal disease to prevent aortic rupture and mortality represents one of the major advances in vascular surgery in the last 20 years. EVAR is associated with lower perioperative morbidity and mortality and has widened the spectrum of patients eligible for repair resulting in improved short-term outcomes. The overall hospital presentations with ruptured and intact aneurysms have declined. As hospital admissions for AAA have decreased, the operative intervention for AAA has also declined due to the reduction in open aortic repair despite the increase in EVAR utilisation. Similar operative trends have also been observed in the UK National Vascular Registry and in the Swedish Vascular Registry. These trends could be explained by the substantial changes that have influenced AAA management over the last two decades including the establishment of EVAR, the implementation of cardiovascular risk factor management, the introduction of statins and increased public health awareness on the importance of lowering blood pressure and smoking cessation. In addition, advances in perioperative medicine and quality improvement initiatives such as centralisation of vascular surgery services may have influenced these trends.
Cardiovascular risk factor management has had a significant impact on the reduction of AAA incidence and prevalence. Pre-existing arterial hypertension increases the risk of developing an AAA significantly. [6] Reports also indicate a dose-dependent relationship between blood pressure and AAA, both for formation and rupture.[7] An analysis of historical and current laboratory data of patients from 12 years before their initial AAA diagnoses found that prior elevated TC, LDL, and triglyceride levels were significantly associated with current AAA. [8]
Line 237: please remove “a had”
This has been noted and removed.
References:
(1) Khashram, M.; Pitama, S.; Williman, J. A.; Jones, G. T.; Roake, J. A. Survival Disparity Following Abdominal Aortic Aneurysm Repair Highlights Inequality in Ethnic and Socio-economic Status. European Journal of Vascular and Endovascular Surgery 2017, 54 (6), 689-696. DOI: 10.1016/j.ejvs.2017.08.018 (acccessed 2023/03/01).
(2) Chiang, N.; Jain, J. K.; Hulme, K. R.; Vasudevan, T. Epidemiology and Outcomes of Abdominal Aortic Aneurysms in New Zealand: A 15-Year Experience at a Regional Hospital. Annals of Vascular Surgery 2018, 46, 274-284. DOI: 10.1016/j.avsg.2017.07.006 (acccessed 2023/03/01).
(3) Sandiford, P.; Mosquera, D.; Bramley, D. Ethnic inequalities in incidence, survival and mortality from abdominal aortic aneurysm in New Zealand. Journal of Epidemiology and Community Health 2012, 66 (12), 1097-1103. DOI: 10.1136/jech-2011-200754.
(4) Khashram, M.; Jones, G. T.; Roake, J. A. Prevalence of abdominal aortic aneurysm (AAA) in a population undergoing computed tomography colonography in Canterbury, New Zealand. Eur J Vasc Endovasc Surg 2015, 50 (2), 199-205. DOI: 10.1016/j.ejvs.2015.04.023 From NLM.
(5) Sandiford, P.; Grey, C.; Salvetto, M.; Hill, A.; Malloy, T.; Cranefield, D.; Bramley, D. The population prevalence of undetected abdominal aortic aneurysm in New Zealand Māori. Journal of Vascular Surgery 2020, 71 (4), 1215-1221. DOI: 10.1016/j.jvs.2019.07.055 (acccessed 2023/03/01).
(6) Kessler, V.; Klopf, J.; Eilenberg, W.; Neumayer, C.; Brostjan, C. AAA Revisited: A Comprehensive Review of Risk Factors, Management, and Hallmarks of Pathogenesis. Biomedicines 2022, 10 (1). DOI: 10.3390/biomedicines10010094 From NLM.
(7) Sweeting, M. J.; Thompson, S. G.; Brown, L. C.; Powell, J. T. Meta-analysis of individual patient data to examine factors affecting growth and rupture of small abdominal aortic aneurysms. Br J Surg 2012, 99 (5), 655-665. DOI: 10.1002/bjs.8707 From NLM.
(8) Wanhainen, A.; Bergqvist, D.; Boman, K.; Nilsson, T. K.; Rutegård, J.; Björck, M. Risk factors associated with abdominal aortic aneurysm: a population-based study with historical and current data. J Vasc Surg 2005, 41 (3), 390-396. DOI: 10.1016/j.jvs.2005.01.002 From NLM.

Reviewer 2 Report
Dear authors,
many thanks for submitting your article entitled "Incidence & Outcomes of Abdominal Aortic Aneurysm Repair in New Zealand from 2001-2021" in the Journal of Clinical Medicine. The manuscript presents the treatment and mortality trends of intact and ruptured AAA in New Zealand across an almost 20-year interval. It contains some interesting data and changing trends. Still, I believe the overall quality of the manuscript can be further improved. Major issues are labeled with asterix.
- Did you register your study in some of the available datasets? I recommend this if you have not done it yet.
* Line 61: You do not have access to the private insurance system datasets? If not please mention this in your limitations.
* Line 67: You miss mortality data from 2018 to 2021. Please mention this in your limitations.
* Line 73, 77-79: It would be valuable if you could add also the following outcomes: all-cause mortality, aortic-related intervention stratified according to the treatment arm, and major adverse cardiovascular events.
- Line 120: What was the overall mortality rate of RAAA patients that underwent invasive treatment?
- Line 130: Please report separately for intact and ruptured AAA.
* Table 1: Can you use inferential statistics and compare these groups? Also, I noticed such high numbers of non-operated RAAA. Can you please report the percentage of palliated, who died before admission, and were denied the surgery?
- Line 142-144: When you mention greater, can you use inferential statistics, not just compare data by observation?
- Line 195,196: please present the crude numbers.
- Line 204-206: can you report mortality separately for EVAR and OR, and their trends?
* How can you explain the fact that despite introducing EVAR in the RAAA setting and other adjuncts, the mortality has not changed? I would add a separate section in the discussion dedicated to this topic.
- Please make a short comment in the conclusion that the mortality rate for RAAA has not changed.
